# Determinants of clinical severity in children with sickle cell disease and confirmed asthma

Gabriel Bafunyembaka[1,2]*, Mathieu Nacher[3], Ariel Makembi[4], Nadia Nathan[5], Narcisse Elenga[6]

1 Department of Pediatrics, Centre Hospitalier de l'Ouest Guyanais (CHOG), Franck Joly Hospital, Saint-Laurent-du-Maroni, French Guiana, France, 2 Department of Pediatrics, General Reference Hospital of Bukavu, Bukavu, Democratic Republic of the Congo, 3 Clinical Investigation Center, Epidemiology/Public Health (Inserm 1424/CIC), Cayenne Hospital, Cayenne, French Guiana, France, 4 Department of Internal Medicine, Centre Hospitalier de l'Ouest Guyanais (CHOG), Franck Joly Hospital, Saint-Laurent-du-Maroni, French Guiana, France, 5 Pediatric Pulmonology Department, Hôpital Trousseau, AP-HP, Sorbonne Université, Paris, France, 6 Department of Pediatrics, Cayenne Hospital, Cayenne, French Guiana, France

* ga.bafunyembaka@chu-guyane.fr

## Abstract

### Background

Asthma is a frequent comorbidity in children with sickle cell disease and has been associated with an increased risk of acute complications, particularly vaso-occlusive crises and acute chest syndrome. However, determinants of clinical severity among children with sickle cell disease and confirmed asthma remain poorly characterized, especially in tropical settings. This study aimed to identify factors associated with clinical severity in this population.

### Methods

We conducted an observational study among children with sickle cell disease followed in French Guiana. The analysis was restricted to children with confirmed asthma. Clinical severity was defined as the occurrence of at least two hospitalizations during the 12 months preceding evaluation for vaso-occlusive crises and/or acute chest syndrome. Factors associated with severity were assessed using univariate and multivariate logistic regression analyses.

### Results

A total of 138 children with sickle cell disease and confirmed asthma were included, of whom 102 (73.9%) presented a severe clinical form. In multivariate analysis, no variable was independently associated with clinical severity. However, a trend toward an increased risk of severe disease was observed among children living in rural areas (adjusted OR = 1.94; 95% CI: 0.77–4.86), while a trend toward a protective effect was observed for *Strongyloides stercoralis* infection (adjusted OR = 0.18; 95%

provided the original author and source are credited.

**Data availability statement:** All relevant data are within the paper and its Supporting Information files.

**Funding:** The author(s) received no specific funding for this work.

**Competing interests:** The authors declare that they have no conflicts of interest.

**Abbreviations:** ACS, Acute Chest Syndrome; CI, Confidence Interval; FEV$_1$, Forced Expiratory Volume in One Second; FVC, Forced Vital Capacity; Hb, Hemoglobin; IQR, Interquartile Range; OR, Odds Ratio; RnIPH, Recherche n'Impliquant Pas la Personne Humaine; VOC, Vaso-Occlusive Crisis.

CI: 0.02–1.51). Allergic sensitization, although frequent (64.5%), was not associated with clinical severity after adjustment (adjusted OR = 0.66; 95% CI: 0.31–1.44).

## Conclusion

Among children with sickle cell disease and confirmed asthma, more than one third experience severe clinical disease. No independent predictors of severity were identified. Observed trends should be interpreted cautiously and considered exploratory. These findings support a stratified approach to sickle cell–associated asthma to identify high-risk children and prevent avoidable acute complications.

## Introduction

Sickle cell disease is one of the most common monogenic disorders worldwide and represents a major public health problem, particularly in children. It is estimated that more than 7 million people are living with sickle cell disease globally, with a high concentration in tropical and subtropical regions [1]. In children, morbidity is mainly driven by acute complications such as vaso-occlusive crises (VOC) and acute chest syndrome (ACS), which are major causes of pediatric hospitalization and mortality [2,3].

Respiratory involvement plays a central role in the pathophysiology and prognosis of sickle cell disease. Acute chest syndrome, in particular, is one of the leading causes of death in children with sickle cell disease and results from complex interactions between pulmonary inflammation, hypoxia, infection, and vascular dysfunction [3,4]. In this context, asthma is now recognized as a frequent comorbidity in children with sickle cell disease and as a potential contributor to increased respiratory morbidity.

Several studies have shown that the presence of asthma in children with sickle cell disease is associated with an increased risk of acute chest syndrome, vaso-occlusive crises, healthcare utilization, and recurrent hospitalizations [5–7]. Seminal studies have notably demonstrated that asthma increases the risk of acute chest syndrome by two- to fourfold in children with sickle cell disease [5,6]. These findings have led to asthma being considered not merely as a comorbidity, but as a true modifier of the clinical phenotype of sickle cell disease.

However, most available studies have focused on the presence or absence of asthma, or on its prevalence, often assessed using clinical criteria alone. More recent work has shown that asthma is frequently underdiagnosed in children with sickle cell disease when based solely on medical history and physical examination, highlighting the value of systematic screening incorporating spirometry [8–10]. Nevertheless, these studies primarily address diagnostic strategies and provide limited insight into the variability of clinical expression among children with sickle cell disease and asthma.

In clinical practice, not all children with sickle cell disease and asthma develop the same complications or exhibit the same degree of disease severity. Some experience

recurrent and severe episodes of vaso-occlusive crises or acute chest syndrome, whereas others have a more favorable course despite confirmed asthma. This heterogeneity suggests the existence of specific determinants of clinical severity within the subgroup of asthmatic children, which remain insufficiently explored.

Furthermore, most available data originate from high-income countries located in temperate regions. In tropical settings, however, additional factors such as environmental exposures, humidity, housing conditions, chronic parasitic infections, and inequalities in access to healthcare may modulate the expression of both asthma and sickle cell disease [11–13]. Helminth infections, in particular, have been associated with modulation of the immune response and may influence the clinical expression of asthma in these regions [14,15].

In this context, it is essential to move beyond the mere diagnosis of asthma and to identify determinants of clinical severity among children with sickle cell disease and confirmed asthma, integrating respiratory, allergological, environmental, and parasitological factors.

The objective of this study was therefore to identify, within a population of children with sickle cell disease and confirmed asthma, the factors associated with a severe clinical form, defined by recurrent acute complications, in a tropical setting.

## Methods

### Study design and population

This was an observational study conducted among children with sickle cell disease followed at the main public hospitals in French Guiana (Cayenne, Kourou, and Saint-Laurent-du-Maroni). Data were collected as part of routine clinical follow-up and a respiratory screening program integrated into standard care.

Among the 390 children with sickle cell disease followed in the participating hospitals, children were screened for asthma. A total of 138 children with confirmed asthma were included in the present analysis. Children younger than 5 years and those with incomplete medical records were excluded.

Clinical severity was assessed over a standardized 12-month follow-up period and defined according to hospitalization frequency. Severe clinical disease was defined as the occurrence of two or more hospitalizations during the follow-up period, whereas non-severe disease corresponded to fewer than two hospitalizations.

To specifically address the study objective, the analysis was deliberately restricted to children with **confirmed asthma**, regardless of the initial diagnostic approach. This strategy aimed to identify determinants of clinical severity within the subgroup of children with sickle cell disease and asthma, rather than to compare asthmatic and non-asthmatic children.

### Follow-up period

The clinical follow-up period corresponded to the 12 months preceding the respiratory assessment performed as part of the asthma screening program. Retrospective data collection was conducted between January 1 and December 31, 2025. Clinical events, including vaso-occlusive crises (VOC) and acute chest syndrome (ACS), were retrospectively collected from hospital medical records, hospitalization reports, and data available in hospital information systems over this standardized period.

This 12-month time window was chosen to ensure homogeneous comparability between patients, to limit bias related to heterogeneous follow-up durations, and to reflect clinically relevant acute morbidity closely related in time to the respiratory assessment.

### Inclusion criteria

Inclusion criteria were as follows:

• age between 5 and 17 years at the time of evaluation;

- confirmed sickle cell disease based on hemoglobin electrophoresis or molecular testing (genotypes SS, SC, $S\beta^0$, or $S\beta^+$);
- confirmed asthma, defined by:
  - Evidence of reversible airflow obstruction on spirometry (increase in $FEV_1 \geq 12\%$ after bronchodilator administration).

**Exclusion criteria**

The following were not included in the analysis:

- children without confirmed asthma;
- children with insufficient clinical or follow-up data to allow assessment of clinical severity over the 12-month period considered.
- children younger than 5 years who were unable to perform spirometry

**Definition of clinical severity**

Clinical severity was pragmatically defined as the occurrence of ≥ 2 hospitalizations **during the 12 months preceding the respiratory assessment** for:

- vaso-occlusive crises (VOC) and/or
- acute chest syndrome (ACS).

This definition was intended to identify children with a high burden of acute morbidity associated with repeated hospital-based healthcare utilization.

The severity variable was coded as a binary outcome:

- 0 = non-severe disease
- 1 = severe disease.

**Variables assessed**

The variables analyzed were grouped into five domains.

**Sociodemographic variables.**

- age (years);
- sex;
- geographic origin, categorized as urban or rural according to place of residence.

**Respiratory variables.**

- baseline spirometry, performed in clinically stable children;
- persistent airflow obstruction, defined as an $FEV_1/FVC$ ratio < 80%;
- bronchodilator reversibility, defined as an increase in $FEV_1 \geq 12\%$.

**Allergological variables.**

- history of allergic rhinitis and eczema;

- results of allergy skin prick tests, considered positive when the wheal diameter was ≥ 3 mm compared with the negative control;

- allergic sensitization, defined as positivity to at least one skin prick test.

## Parasitological variables

- results of stool parasitological examinations;

- presence or absence of intestinal parasites, with particular attention to *Strongyloides stercoralis*. Stool samples were collected and analyzed in certified hospital laboratories using standard parasitological methods, including direct microscopic examination and concentration techniques. Strongyloides stercoralis infection was defined by the identification of larvae on stool microscopy. No serological confirmation was available in the present study.

## Sickle cell disease–related variables

- Sickle cell genotypes were categorized into severe genotypes (HbSS and HbSβ⁰) and milder genotypes (HbSC and HbSβ⁺) for subgroup analysis.

- treatment with hydroxyurea (yes/no) at the time of evaluation;

- documented history of acute chest syndrome.

## Statistical analysis

Statistical analyses were performed exclusively in children with confirmed asthma.

Quantitative variables were described using mean and standard deviation or median and interquartile range (IQR), depending on their distribution. Categorical variables were described using counts and percentages.

Comparisons between children with severe clinical disease and those with non-severe disease were performed using:

- the Mann–Whitney U test for non-normally distributed quantitative variables;

- the chi-square test or Fisher's exact test, as appropriate, for categorical variables.

- Model goodness-of-fit was assessed using the Hosmer–Lemeshow test.

Factors associated with clinical severity were explored using univariate logistic regression. Variables associated with a p value < 0.20 in univariate analyses, as well as variables considered clinically relevant, were included in a multivariate logistic regression model to identify factors independently associated with severity. To reduce the risk of model overfitting, the number of variables included in the multivariate model was restricted according to the number of outcome events, following recommended statistical practices.

Results were expressed as adjusted odds ratios (ORs) with their 95% confidence intervals (95% CIs). Statistical significance was defined as a p value < 0.05.

## Ethical considerations

This study was conducted in accordance with French regulations governing research not involving human participants (Recherche n'Impliquant Pas la Personne Humaine, RnIPH). Data were retrospectively collected from routine clinical records and pseudonymized prior to analysis. Patients and their legal guardians were informed of the possible use of anonymized data for research purposes and did not express opposition. The authors did not have access to information that could identify individual participants during or after data collection.

## Results

Among the 138 children included in the analysis, 102 children (73.9%) presented severe clinical disease (≥ 2 hospitalizations), while 36 children (26.1%) were classified as non-severe (<2 hospitalizations). The median age was 8.0 years (IQR: 6.0–11.0), and 45.7% were male. The majority of children had the SS genotype (75.4%) and resided in rural areas (73.9%). The flow chart of participants included in the study is presented in Fig 1.

Treatment with hydroxyurea was ongoing in 77.4% of patients. From a respiratory standpoint, persistent airflow obstruction, defined as an $FEV_1$/FVC ratio<80%, was observed in 58.4% of children, whereas 20.4% had normal spirometry. Allergic sensitization, defined by positivity to at least one skin prick test, was identified in 64.5% of children. Intestinal parasites were detected in 35.8% of patients, including *Strongyloides stercoralis* in 8.8%. Detailed population characteristics are presented in Table 1.

A severe clinical expression, defined as the occurrence of at least two hospitalizations during the 12 months preceding evaluation for vaso-occlusive crises and/or acute chest syndrome, was observed in 102 children, corresponding to 73.9% of the study population. This high proportion illustrates the heterogeneity of clinical expression among children with sickle

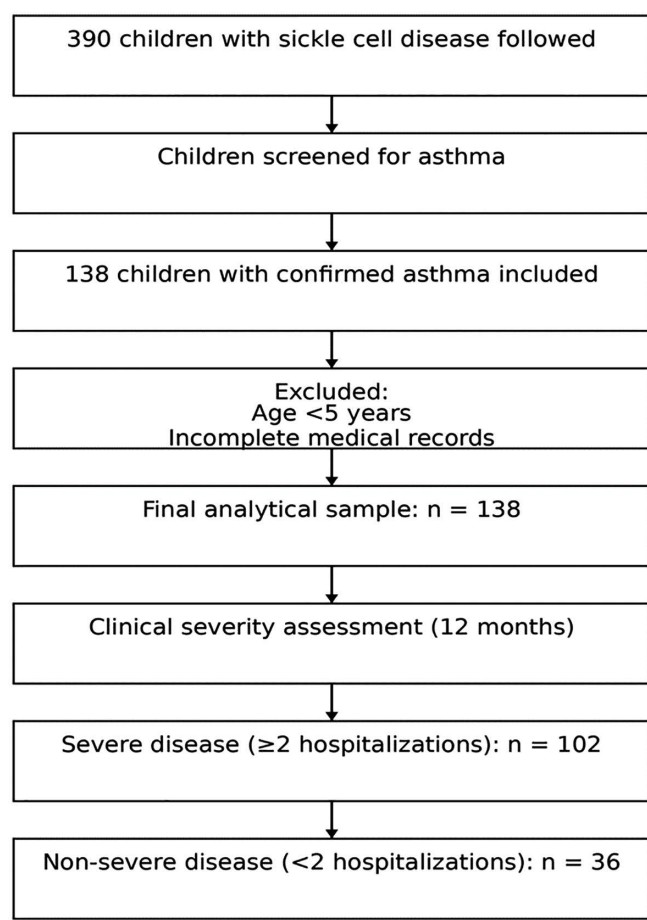

**Fig 1. Flow chart of the study population.** Among 390 children with sickle cell disease followed in participating hospitals, 138 children with confirmed asthma were included after exclusion of children younger than 5 years and those with incomplete medical records. Clinical severity was assessed over a 12-month follow-up period and classified as severe (≥2 hospitalizations) or non-severe (<2 hospitalizations).

**Table 1. Characteristics of children with sickle cell disease (study population).**

| Variables | Value |
|---|---|
| Confirmed asthma, n (%) | 138 (100) |
| Clinical severity (≥2 hospitalizations), n (%) | 102 (73.9) |
| Age, median (IQR), years | 8.0 (6.0–11.0) |
| Male sex, n (%) | 63 (45.7) |
| SS genotype, n (%) | 106 (76.8) |
| SC genotype, n (%) | 32 (23.2) |
| Rural residence, n (%) | 102 (73.9) |
| Hydroxyurea treatment, n (%) | 106 (76.8) |
| Persistent airflow obstruction (FEV$_1$/FVC<80%), n (%) | 80 (58.0) |
| Normal spirometry, n (%) | 28 (20.3) |
| Allergic sensitization (≥1 positive skin test), n (%) | 89 (64.5) |
| Intestinal parasites present, n (%) | 49 (35.5) |
| Strongyloides stercoralis positive, n (%) | 12 (8.7) |

**Clinical severity defined as** ≥2 hospitalizations **per year for vaso-occlusive crises and/or acute chest syndrome.**

cell disease and confirmed asthma. A subgroup analysis based on genotype severity was performed. Severe genotypes were defined as HbSS and HbSβ$^o$, while milder genotypes included HbSC and HbSβ$^+$

Severe clinical outcomes were more frequent among children with HbSS genotype compared with those with HbSC genotype (77.4% vs 57.1%). This association was statistically significant (p=0.0019).

In univariate analysis, clinical severity was more frequent among children living in rural areas than among those living in urban areas (**59.8% vs 40.2%**) although this difference did not reach statistical significance (p=0.083). The presence of *Strongyloides stercoralis* was less frequent among children with severe disease compared with those with non-severe disease (2.0% vs. 12.5%), suggesting a trend toward a protective effect (p=0.055). No significant associations were observed between clinical severity and age, sex, sickle cell genotype, hydroxyurea treatment, persistent airflow obstruction, normal spirometry, allergic sensitization, or the overall presence of intestinal parasites.

In multivariate logistic regression analysis, no variable was independently associated with clinical severity. Nevertheless, **exploratory trends were observed** after adjustment. Residence in rural areas was associated with an increased risk of severe disease (adjusted OR = 1.94; 95% CI: 0.77–4.86), whereas positivity for *Strongyloides stercoralis* was associated with a reduced risk of severity (adjusted OR = 0.18; 95% CI: 0.02–1.51). In contrast, allergic sensitization (adjusted OR = 0.66; 95% CI: 0.31–1.44) and persistent airflow obstruction (adjusted OR = 0.89; 95% CI: 0.41–1.95) were not associated with clinical severity after adjustment (Tables 2 and 3).

## Discussion

### Main findings

In this study focusing exclusively on children with sickle cell disease and confirmed asthma, more than two thirds (73.9%) presented a severe clinical form, defined by recurrent hospitalizations for vaso-occlusive crises (VOC) and/or acute chest syndrome (ACS). This high proportion highlights that, even within a diagnostically homogeneous asthmatic population, the clinical expression of sickle cell disease remains highly heterogeneous, which is consistent with the wide variability in respiratory morbidity described in pediatric sickle cell disease [1–4,16–18].

Analysis of our data showed that allergic sensitization, although frequent, was not associated with clinical severity. This finding suggests that severe sickle cell–associated asthma is not primarily driven by classical allergic mechanisms,

**Table 2. Factors associated with clinical severity – Univariate analysis.**

| Variable | Severe (n = 102) | Non-severe (n = 36) | p value |
|---|---|---|---|
| Rural residence, n (%) | 61 (59.8) | 41 (40.2) | 0.083 |
| Strongyloides stercoralis positive, n (%) | 1 (1.0) | 11 (30.6) | 0.055 |
| Allergic sensitization, n (%) | 57 (55.9) | 32 (88.9) | >0.05 |
| Persistent airflow obstruction, n (%) | 51 (50.0) | 29 (80.6) | >0.05 |

**Table 3. Factors associated with clinical severity – Multivariate logistic regression.**

| Variable | Adjusted OR | 95% CI | p value |
|---|---|---|---|
| Rural residence | 1.94 | 0.77–4.86 | 0.15 |
| Strongyloides stercoralis positive | 0.18 | 0.02–1.51 | 0.11 |
| Allergic sensitization | 0.66 | 0.31–1.44 | 0.30 |
| Persistent airflow obstruction | 0.89 | 0.41–1.95 | 0.77 |

in contrast to asthma in the general pediatric population [13]. Indeed, pathophysiological studies have shown that airway hyperresponsiveness and airway inflammation observed in children with sickle cell disease may occur independently of atopy, through mechanisms related to chronic hypoxia and systemic inflammation [19,20].

In contrast, exploratory trends were observed for certain contextual factors. Residence in rural areas was associated with an increased risk of severe disease, whereas positivity for Strongyloides stercoralis was associated with a reduced risk. The absence of a single factor explaining clinical severity supports the multifactorial nature of severe sickle cell–associated asthma, as illustrated in our conceptual framework and previously suggested by studies integrating respiratory, vascular, and environmental factors [9,16,21].

## Comparison with international data

The association between asthma and acute complications of sickle cell disease is well established. Several North American and Caribbean studies have demonstrated that asthma is associated with a significantly increased risk of acute chest syndrome, vaso-occlusive crises, and hospitalizations in children with sickle cell disease [5–7,22,23]. Some studies have also reported an association between asthma and increased mortality in patients with sickle cell disease, underscoring the prognostic impact of this comorbidity [24].

However, most of these studies compared asthmatic and non-asthmatic children without examining variability in clinical severity within the asthmatic subgroup. Our findings complement these data by showing that not all children with sickle cell disease and confirmed asthma experience the same degree of clinical severity. This intra-group heterogeneity aligns with more recent observations suggesting the existence of distinct respiratory phenotypes in sickle cell disease, with different pulmonary functional trajectories throughout childhood and adolescence [8,25].

Unlike asthma in the general pediatric population, in which atopy is a central determinant of disease severity and poor control [13], allergic sensitization was not associated with clinical severity in our population. This dissociation between allergy and clinical severity is consistent with studies conducted in tropical and subtropical settings, where the clinical expression of asthma appears to be less dependent on atopy and more strongly influenced by environmental and socio-economic factors [11,13].

The trend toward an association between rural residence and clinical severity observed in our study is also consistent with international data highlighting the role of socio-environmental determinants in sickle cell disease–related morbidity [11,12]. Previous studies have shown that access to healthcare, housing conditions, and exposure to humid or polluted

environments may modulate respiratory expression and the frequency of acute complications in children with sickle cell disease [16,26,27].

A particularly original finding of our study is the trend toward a protective effect of *Strongyloides stercoralis*. Experimental and epidemiological studies have demonstrated that certain helminth infections can modulate Th2-type immune responses and attenuate the clinical expression of asthma [14,15]. This hypothesis is further supported by data suggesting that parasite-induced immune modulation may influence airway hyperresponsiveness and airway inflammation [28]. Although this association did not reach statistical significance after adjustment, it opens original perspectives for understanding sickle cell–associated asthma in tropical settings.

Finally, the absence of an independent association between persistent airflow obstruction and clinical severity after adjustment suggests that isolated respiratory function is insufficient to predict acute complications in children with sickle cell disease. Imaging and pulmonary function studies have shown that respiratory abnormalities in sickle cell disease are often dissociated from acute clinical events, implicating systemic vascular and inflammatory mechanisms [17,21,29,30,31].

### Strengths and limitations

The main strength of this study lies in its targeted approach, restricted to children with sickle cell disease and confirmed asthma, which allowed exploration of determinants of clinical severity within the asthmatic subgroup itself, in a tropical setting that remains underrepresented in the literature. The availability of detailed data on respiratory function, allergy, and parasitology represents an important additional strength.

Limitations include the observational design of the study and the sample size, which may have limited statistical power to detect independent associations, particularly for parasitological factors. These findings should therefore be interpreted as hypothesis-generating and warrant confirmation in larger cohorts. The definition of severity based on hospitalization frequency may partially reflect healthcare utilization patterns rather than disease severity alone, particularly in settings where access to care varies geographically. In addition, biological markers such as peripheral blood eosinophil counts and total IgE levels were not systematically available. These parameters could have improved the characterization of allergic and parasitic profiles and may have identified additional subclinical infections.

### Conclusion

No independent predictors of clinical severity were identified in this study. Observed trends should be interpreted cautiously and considered exploratory. These findings generate hypotheses for future prospective studies aimed at better identifying determinants of severity in children with sickle cell disease and asthma.

### Supporting information

**S1 Table. Individual-level characteristics of children with sickle cell disease included in the respiratory assessment.** The table summarizes demographic data, clinical severity indicators, asthma screening results (ISAAC questionnaire and spirometry), allergological testing, and stool parasitological findings. Spirometric obstruction was defined as $FEV_1/FVC < 80\%$, and bronchodilator reversibility as an improvement in $FEV_1 \geq 12\%$. This table provides the detailed dataset underlying the descriptive and analytical results presented in the manuscript.
(XLSX)

### Author contributions

**Conceptualization:** Narcisse Elenga.

**Data curation:** Gabriel Bafunyembaka, Ariel Makembi.

**Investigation:** Ariel Makembi.

**Methodology:** Nadia Nathan, Mathieu Nacher.

**Project administration:** Narcisse Elenga.

**Software:** Gabriel Bafunyembaka.

**Supervision:** Narcisse Elenga.

**Validation:** Nadia Nathan, Mathieu Nacher, Narcisse Elenga.

**Writing – original draft:** Gabriel Bafunyembaka.

**Writing – review & editing:** Nadia Nathan, Mathieu Nacher, Ariel Makembi.

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
