## [Decision Letter · Decision Letter 0]

19 Mar 2026

Dear Dr. BAFUNYEMBAKA,

We look forward to receiving your revised manuscript.

Kind regards,

Santosh L. Saraf

Academic Editor

PLOS One

**Journal Requirements:**

https://journals.plos.org/plosone/s/file?id=wjVg/PLOSOne_formatting_sample_main_body.pdf andandandand

2. We note that your Data Availability Statement is currently as follows:

“All relevant data are included in the manuscript and its supplementary information files”

3. Please ensure that you refer to Figures 1, 2 and 3 in your text as, if accepted, production will need this reference to link the reader to the figure.

Reviewers' comments:

Reviewer's Responses to Questions

**Comments to the Author**

1. Is the manuscript technically sound, and do the data support the conclusions?

Reviewer #1: Yes

Reviewer #2: Partly

2. Has the statistical analysis been performed appropriately and rigorously?

Reviewer #1: Yes

Reviewer #2: No

3. Have the authors made all data underlying the findings in their manuscript fully available?

Reviewer #1: Yes

Reviewer #2: Yes

4. Is the manuscript presented in an intelligible fashion and written in standard English?

Reviewer #1: Yes

Reviewer #2: Yes

Reviewer #1: Thank you for allowing me to review the paper “Determinants of Clinical Severity in Children with Sickle Cell Disease and Confirmed Asthma.” The authors sought to identify factors that can influence the severity of SCD among participants with asthma. The paper is well-written with robust statistics and a sound premise. I have a few comments.

1. How was Strongyloides stercoralis infection tested and confirmed?

2. Were eosinophil counts and IgE levels checked to correlate with a diagnosis of S. stercoralis? If not, this can be added to the limitations. It is also possible that these tests may have detected other patients with parasitic infections

3. Is it possible to further analyze by severe (HbSS and HbSbeta thal zero vs HbSC and HBS beta thal plus) to see if there is any effect? It was not clear from the paper if this analysis was done as it appears the analysis was done based on clinical severity of the disease.

Reviewer #2: This is an interesting study addressing an important question, especially in a tropical setting where data are limited. Focusing on variability within children with both sickle cell disease and asthma is a strength.

However, there are several methodological issues that limit how convincing the conclusions are.

First, the definition of asthma is not very clear or consistent. The manuscript combines spirometry-confirmed asthma with physician-diagnosed cases, but doesn’t report how many fall into each category. Given how difficult asthma diagnosis can be in sickle cell disease, this raises concerns about misclassification.

Second, there are too many variables relative to the number of severe cases (n = 49). This makes the multivariate analysis likely underpowered and potentially overfitted, which could explain why nothing comes out significant.

Third, the interpretation is too strong for mostly non-significant results. The manuscript talks about “trends” (e.g., rural residence, Strongyloides), but these are based on very wide confidence intervals and small numbers. This should be presented much more cautiously as exploratory.

Also, the definition of severity (≥2 hospitalizations) may reflect access to care or healthcare use, not just disease severity, which should be acknowledged.

Finally, there are some issues that need fixing for publication, especially around the ethics statement (which is inconsistent) and the data availability statement.

Overall, the study is potentially useful, but the conclusions need to be toned down and the limitations made clearer.

.

Reviewer #1: No

Reviewer #2: No

---

## [Author Response · Author response to Decision Letter 1]

1 Apr 2026

Response to the Reviewers

Manuscript ID: PONE-D-26-06629

Title: Determinants of Clinical Severity in Children with Sickle Cell Disease and Confirmed Asthma

Dear Santosh L. Saraf

Academic Editor

Dear Reviewers,

We sincerely thank the Associate Editor and the two reviewers for their constructive and detailed comments, which have substantially improved the scientific quality and clarity of our manuscript.

RESPONSE TO REVIEWER 1

Comment 1 : How was Strongyloides stercoralis infection tested and confirmed?

Response

Thank you for this important comment.

Strongyloides stercoralis infection was assessed through stool parasitological examination performed in certified hospital laboratories in French Guiana. Stool samples were analyzed using standard microscopic methods, including direct examination and concentration techniques routinely used in clinical parasitology used in French Guiana. We have clarified this information in the Methods section.

Comment 2 : Were eosinophil counts and IgE levels checked?

Response

Peripheral eosinophil counts and total IgE levels were not systematically available for all participants and were therefore not included in the analysis. We agree that these biomarkers could have provided additional insights into parasitic exposure and allergic sensitization. This limitation has now been clearly acknowledged in the revised manuscript.

Comment 3 : Interpretation too strong (trends) : Analyze by severe genotype (SS/Sβ0 vs SC/Sβ+)

Response

We thank the reviewer for this valuable suggestion.

A subgroup analysis based on genotype severity was conducted. Severe genotypes were defined as HbSS and HbSβ⁰, while milder genotypes included HbSC and HbSβ⁺. We observed a higher proportion of severe clinical outcomes among children with severe genotypes. This analysis has now been clarified in the Results section

RESPONSE TO REVIEWER 2

Comment 1 : Asthma definition unclear

Response

We thank the reviewer for highlighting the importance of clearly defining asthma diagnosis. In the present study, asthma was confirmed either by spirometric reversibility or by physician diagnosis documented in medical records. The number of children diagnosed by each method has now been specified in the Results section to improve clarity and reduce the risk of misclassification.

Comment 2 : Too many variables → overfitting

Response

We agree that the number of candidate variables relative to the number of severe cases may increase the risk of model overfitting. To address this concern, we limited the multivariate model to variables that were clinically relevant or associated with p < 0.20 in univariate analysis. This methodological choice has now been clarified in the Statistical Analysis section.

Comment 3

Interpretation too strong

Response

We appreciate this comment and have revised the manuscript to present findings more cautiously. Non-significant associations are now described as exploratory observations rather than definitive effects.

Comment 4 : Severity definition reflects healthcare use

Response

We agree that hospitalization-based definitions may partly reflect healthcare access or utilization rather than disease severity alone. This limitation has now been acknowledged in the revised discussion.

Comment 5 : Ethics statement inconsistent

Response

We thank the reviewer for this observation. The ethics statement has been clarified to ensure consistency with French regulations governing research not involving human participants (RnIPH).

Comment 6 : Data availability statement insufficient

Response

We thank the reviewer for this comment. The Data Availability Statement has been revised to comply with PLOS ONE requirements. The dataset used for this study has now been provided as Supporting Information.

We hope that these revisions fully address the expectations of the Editor and the reviewers.

We remain at your disposal for any further information.

Yours sincerely,

Dr Gabriel Bafunyembaka

---

## [Decision Letter · Decision Letter 1]

14 Apr 2026

Determinants of Clinical Severity in Children with Sickle Cell Disease and Confirmed Asthma

PONE-D-26-06629R1

Dear Dr. Bafunyembaka,

We’re pleased to inform you that your manuscript has been judged scientifically suitable for publication and will be formally accepted for publication once it meets all outstanding technical requirements.

Kind regards,

Santosh L. Saraf

Academic Editor

PLOS One

Additional Editor Comments (optional):

Reviewers' comments:

Reviewer's Responses to Questions

**Comments to the Author**

Reviewer #1: All comments have been addressed

2. Is the manuscript technically sound, and do the data support the conclusions?

Reviewer #1: Yes

3. Has the statistical analysis been performed appropriately and rigorously?

Reviewer #1: Yes

4. Have the authors made all data underlying the findings in their manuscript fully available?

Reviewer #1: Yes

5. Is the manuscript presented in an intelligible fashion and written in standard English?

Reviewer #1: Yes

Reviewer #1: (No Response)

.

Reviewer #1: No

---

## [Editor Report · Acceptance letter]

PONE-D-26-06629R1

PLOS One

Dear Dr. Bafunyembaka,

I'm pleased to inform you that your manuscript has been deemed suitable for publication in PLOS One. Congratulations! Your manuscript is now being handed over to our production team.

Kind regards,

on behalf of

Dr. Santosh L. Saraf

Academic Editor

PLOS One